# Exploring Sex Differences of Beta-Blockers in the Treatment of Hypertension: A Systematic Review and Meta-Analysis

**DOI:** 10.3390/biomedicines11051494

**Published:** 2023-05-22

**Authors:** Nick Wilmes, Eveline M. van Luik, Esmée W. P. Vaes, Maud A. M. Vesseur, Sophie A. J. S. Laven, Zenab Mohseni-Alsalhi, Daniek A. M. Meijs, Cédric J. R. Dikovec, Sander de Haas, Marc E. A. Spaanderman, Chahinda Ghossein-Doha

**Affiliations:** 1Department of Obstetrics and Gynaecology, Maastricht University Medical Center (MUMC+), 6229 ER Maastricht, The Netherlands; 2Cardiovascular Research Institute Maastricht, School for Cardiovascular Diseases, Maastricht University, 6229 ER Maastricht, The Netherlands; 3GROW—School for Oncology and Developmental Biology, Maastricht University, 6229 ER Maastricht, The Netherlands; 4Department of Obstetrics and Gynaecology, Radboud University Medical Center, 6525 GA Nijmegen, The Netherlands; 5Department of Cardiology, Maastricht University Medical Center (MUMC+), 6229 ER Maastricht, The Netherlands

**Keywords:** hypertension, beta-blockers, sex differences, systematic review, meta-analysis

## Abstract

Aims: In the prevention of cardiovascular morbidity and mortality, early recognition and adequate treatment of hypertension are of leading importance. However, the efficacy of antihypertensives may be depending on sex disparities. Our objective was to evaluate and quantify the sex-diverse effects of beta-blockers (BB) on hypertension and cardiac function. We focussed on comparing hypertensive female versus male individuals. Methods and results: A systematic search was performed for studies on BBs from inception to May 2020. A total of 66 studies were included that contained baseline and follow up measurements on blood pressure (BP), heart rate (HR), and cardiac function. Data also had to be stratified for sex. Mean differences were calculated using a random-effects model. In females as compared to males, BB treatment decreased systolic BP 11.1 mmHg (95% CI, −14.5; −7.8) vs. 11.1 mmHg (95% CI, −14.0; −8.2), diastolic BP 8.0 mmHg (95% CI, −10.6; −5.3) vs. 8.0 mmHg (95% CI, −10.1; −6.0), and HR 10.8 beats per minute (bpm) (95% CI, −17.4; −4.2) vs. 9.8 bpm (95% CI, −11.1; −8.4)), respectively, in both sexes’ absolute and relative changes comparably. Left ventricular ejection fraction increased only in males (3.7% (95% CI, 0.6; 6.9)). Changes in left ventricular mass and cardiac output (CO) were only reported in males and changed −20.6 g (95% CI, −56.3; 15.1) and −0.1 L (95% CI, −0.5; 0.2), respectively. Conclusions: BBs comparably lowered BP and HR in both sexes. The lack of change in CO in males suggests that the reduction in BP is primarily due to a decrease in vascular resistance. Furthermore, females were underrepresented compared to males. We recommend that future research should include more females and sex-stratified data when researching the treatment effects of antihypertensives.

## 1. Introduction

Cardiovascular disease (CVD) affects over one billion people globally [1,2]. Hypertension is the most significant contributor to mortality in CVD and is, therefore, the most important and modifiable risk factor to be targeted [3,4]. In CVD, mortality rates differ between sexes [5]. Although hypertension and its CVD risks occur less before menopause compared to age-matched males; after menopause, females rapidly catch up, ultimately exceeding males [6,7].

Weight control and antihypertensive drugs are amongst the most pivotal in controlling hypertension [8]. There are differences suggested in females compared to males in system biology, clinical manifestations, treatment effects, and outcomes of CVD [9,10]. Differences in sex chromosomes and hormones, biological variations in drug pharmacokinetics and pharmacodynamics, female-specific diseases such as gestational, cardiovascular, and cardiometabolic diseases, and differences in clinical expression or recognition resulting from the underrepresentation of females in clinical trials may all contribute to the disparities observed between males and females [11,12,13,14,15]. Treatment effects may consequently differ between females and males, even though for both sexes beta-blockers (BB) are considered amongst the first-line drug treatment options in current guidelines, together with angiotensin-converting enzyme inhibitors (ACE-I), angiotensin receptor blockers (ARB), calcium channel blockers (CCB), and diuretics [16].

This raises the question as to whether BBs are equally effective in controlling hypertension in both males and females. Therefore, in this systematic review and meta-analysis, we study the effects of BB treatment on hemodynamic variables in female versus male adults.

## 2. Materials and Methods

### 2.1. Series of Meta-Analysis

The search, inclusion, and exclusion criteria are developed for a series of systematic reviews and meta-analyses. The goal of this series is to assess the effect of the five major groups of antihypertensive drugs on cardiovascular outcomes in females as compared to males. This systematic review and meta-analysis investigates the effect of BBs. Our review was registered in Prospero database with registration number: CRD42021273583.

### 2.2. Literature Search

A literature search from inception (1945) up to May 2020 was conducted in PubMed (NCBI) and EMBASE (Ovid) for studies evaluating the effects of antihypertensive medication on cardiovascular and hemodynamic variables in hypertensive individuals. The search strategy focused on cardiac geometry, heart failure, diastolic dysfunction, myocardial infarction (MI), and cerebral vascular accident (CVA) as detailed in the Appendix A. The search limits used were ‘humans’ and ‘journal article’. The search served to study two objectives:-To determine the representation of females in studies on the effect of antihypertensive drugs on CVD for the past century.-To study differences and similarities between males and females on the effect of antihypertensive medication on cardiac function and structure.

### 2.3. Eligibility Criteria

Studies had to focus on acute (0–14 days), semi-acute (15–30 days), and/or chronic (>31 days) therapy with at least one class of antihypertensives (BBs, ACE-I, ARBs, CCBs, and diuretics) in male and/or female adults (≥18 years). 

Moreover, studies had to include a mean with a standard deviation (SD), standard error (SE), or a 95% confidence interval (95% CI) of baseline and follow up measurements of the variables of interest. Variables of primary interest were systolic and diastolic blood pressure (SBP and DBP), mean arterial blood pressure (MAP), and heart rate (HR). Variables of secondary interest were cardiac output (CO), left ventricular ejection fraction (LVEF), and left ventricular mass (LVM). Studies also had to report the mean dose or the dose range, as well as treatment duration. Finally, the antihypertensive treatment had to be compared to a reference group (control, placebo, or another antihypertensive medication group). Mean values with SD were requested from the authors by email if articles presented their data differently. All study designs which reported baseline and follow up measurement were included in this systematic review.

### 2.4. Study Selection

After the initial search, studies were screened based on title and abstract. During this selection, other systematic reviews and meta-analyses, literature reviews, case reports, animal studies, and in vitro studies were excluded. Moreover, articles in other languages than English and Dutch were excluded. The remaining studies were screened for eligibility based on their full text using the eligibility criteria. Studies were excluded if they did not separate outcomes by antihypertensive medication (if participants received more than one antihypertensive medication as an intervention) or did not report the treatment duration and a mean dose or dose range for the antihypertensive medication. Studies with patients undergoing invasive operations, patients who engaged in exercise during measurements, or patients undergoing dialysis or chemotherapy were excluded as well. In case studies that did not stratify outcomes by sex and all other eligibility criteria were met, authors from articles published in 1980 and later were sent an email or were approached via research gate to request sex-specific data. A reminder was sent after two weeks if authors did not reply. Email addresses from either the first author, corresponding author, or head of the department were retrieved from the article, research gate, or using their name and/or institution to search the internet. If no contact details were found or if authors did not respond within three weeks after being sent a reminder, the article was excluded from this systematic review. The reason for exclusion was registered during the full-text selection. Both selection steps were performed in pairs in a blinded, standardised manner (title–abstract pairs: MA-EV, CD-SL, EL-DM, ZM-JW, and MV-NW; full-text pairs: CD-NW, EL-MV, DM-SL, and EV-JW). Extractions were performed independently by two investigators. Discrepancies were resolved by dialogue or discussion with a third independent investigator.

### 2.5. Data Extraction

Study characteristics (sample size, control group, and study design), anthropometric data (age and ethnicity), intervention characteristics (dose, duration, and method of measurement), and effect measures (mean and SD at baseline and after beta blockage intervention of SBP, DBP, MAP, HR, CO, LVEF, and LVM) were collected in a predesigned format made by the investigators. The study results were separately extracted for males and females. In this systematic review, only blood pressure (BP) data measured using non-invasive methods was extracted. For the other variables, multiple methods were allowed. Moreover, studies had to report a mean at baseline and post-intervention mean including SD for the outcome variables. Studies that only reported a change from baseline without post-intervention mean and SD were not included. Data extraction was performed by two investigators (RA and LK). 

### 2.6. Quality Assessment

The included studies were assessed for quality and risk of bias using the Cochrane recommended Risk of Bias 2 (RoB2) tool [17]. Studies were scored with “Low risk of bias”, “Some concerns”, or “High risk of bias” on five domains including randomisation process, deviations from intended interventions, missing data, outcome measurements, and data reporting. To receive an overall risk of bias judgement of “Low risk of bias”, all domains had to receive this judgement. To receive an overall judgment of “High risk of bias”, at least one of the domains was scored as such. All other domain score combinations would rate a study with an overall judgement of “Some concerns”. The quality assessment was performed by two reviewers (RA and LK) and differences were solved by a third independent reviewer (DM and SL).

### 2.7. Statistical Analysis

If the SE or 95% CI was reported in the article, the SD was calculated according to the Cochrane Handbook for Systematic Review of Interventions [18]. Changes in the cardiovascular and haemodynamic variables from baseline were separately analysed for males and females using a random-effects model as described by DerSimonian and Laird [19]. Because the included studies had some variation in study population and design, the random-effects model was chosen to account for this interstudy variation [20]. Egger’s regression test for funnel plot asymmetry was conducted to test for publication bias for each cardiovascular variable [21]. The primary outcome was the mean difference and 95% CI between baseline and follow up of the intervention, visualised in forest plots. The relative change from baseline in percentage including 95% CI was also calculated and reported in parentheses behind the mean difference in the text. The I2 statistic, the ratio between heterogeneity and variability, was calculated as a measure of consistency and expressed as percentage in the forest plots. I2 is able to distinguish heterogeneity in data from solely sampling variance [20]. Interpretation of I2 was based on the guidelines in the Cochrane Handbook for Systematic Review of Interventions [20]. Sources of clinical heterogeneity (compound, treatment duration, and dosage) and methodological heterogeneity (quality of study) were investigated by meta-regression analyses using a mixed-effects model [20]. For the meta-analyses and meta-regression analyses, the meta package in the statistical programme R version 4.0.3. was used [22,23]. 

## 3. Results

### 3.1. Study Selection

The literature search in PubMed and Embase provided a total of 73,867 unique records after removing duplicates (Figure 1). During the first screening, 58,737 articles were excluded resulting in 15,130 articles that were assessed based on the full text. Of those articles, 14,916 met at least one exclusion criterium and were excluded. For 766 articles (5%), it was not possible to find or access the full text at the university library or online. A total of 1141 articles (8%) had an unsuitable study design. This criterium was met when for example only measurements were taken during exercise, or SBP and DBP were measured intra-arterial. A total of 1058 articles (7%) did not report original research data; these articles were reviews for example. In 1886 articles (13%), no antihypertensives were given to the patients participating. In 2141 articles (14%), antihypertensives were given but treatment results were not stratified by antihypertensive. A total of 1949 articles (13%) were excluded because treatment results were not stratified by sex. A total of 153 articles (1%) did not have reference measurements. A total of 3864 articles (26%) did not contain any measurements of interest. In 536 articles (4%), data were not suitably reported. In 984 articles (6%), there was no information provided regarding either dose, duration, or both. Finally, 438 articles (3%) were excluded because of other complications. At the end of the selection procedure, a total of 214 articles were classified as suitable for inclusion (Figure 1). Eventually, in 63 of those articles, BBs were administered as treatment.

### 3.2. Study Characteristics

The total number of participants included in this meta-analysis was 2052, of whom 414 (20.2%) were females. Table 1 contains the study characteristics and anthropomorphic data of the participants. The mean age of the participants of all the included studies was 52.1 ± 12.8 years. A total of 18 studies reported on metoprolol [24,25,26,27,28,29,30,31,32,33,34,35,36,37,38,39,40,41,42], 11 on atenolol [39,43,44,45,46,47,48,49,50,51,52], ten on propranolol [53,54,55,56,57,58,59,60,61,62], nine on carvedilol [46,63,64,65,66,67,68,69,70], and four on acebutolol [43,71,72], labetalol [59,73,74,75], and nebivolol [32,76,77,78]. Three studies wrote about bucindolol [30,79,80] and two about pindolol [58,81]. Each of the following BBs had one study included; bisoprolol [82], celiprolol [37], dilevalol [83], indenolol [84], nadolol [61], nipradolol [56], timolol [85], and tolamolol [53]. A total of 31 studies reported on SBP [24,26,27,28,29,32,34,35,38,41,42,44,45,48,50,54,55,56,57,61,62,63,64,72,73,78,79,82,83,84,86], 28 on DBP [24,26,27,28,32,35,38,42,44,45,46,48,49,50,54,55,56,57,62,63,64,72,73,78,79,82,83,84], seven on MAP [32,42,48,52,63,78,86], 55 on HR [24,25,26,27,28,29,30,31,32,33,34,35,36,37,38,39,41,42,43,44,45,46,47,48,49,50,51,53,54,55,56,57,58,59,60,61,63,64,65,66,69,70,71,73,74,75,76,78,79,80,81,82,83,84,85], five on CO [27,28,66,76,80], 23 on LVEF [25,27,29,31,32,33,34,40,41,42,45,48,54,63,65,66,67,68,69,72,73,77,84], and 4 on LVM [29,44,45,50]. A total of 23 studies measured the acute effects of BBs [24,27,33,34,36,37,43,46,47,51,53,55,56,58,59,60,71,74,75,76,79,81,84], four studies measured the sub-acute effects of BBs [54,61,73,82], and the remaining 36 reported on the chronic effects [25,26,28,29,30,31,32,35,38,39,40,41,42,44,45,48,49,50,52,57,62,63,64,65,66,67,68,69,70,72,77,78,80,83,85,86]. The study types of the included articles consisted of 26 RCTs [26,27,28,29,30,32,35,37,43,44,45,46,48,50,51,53,55,57,62,64,68,70,71,78,79], 34 prospective cohort studies [24,25,31,33,34,36,39,40,41,42,47,49,54,56,58,59,60,61,63,65,66,67,69,73,74,75,76,77,80,81,82,83,84,85], two retrospective studies [72,86], and one cross-sectional study [52]. 

A total of 49 studies included only male subjects [24,25,27,28,29,30,31,33,34,36,37,42,43,44,45,46,47,49,50,51,52,53,55,56,57,58,59,60,61,62,63,64,66,67,69,71,72,73,74,75,76,79,80,81,82,83,84,85,86], no studies included only female subjects, and 14 studies included both male and female subjects [26,32,35,38,39,40,41,48,54,65,68,70,77,78]. For six of those 14 studies, sex-stratified data were not provided in the original article [32,35,38,48,77,78]. Therefore, they had to be inquired from the authors of these studies, whereafter they could be included in this meta-analysis. Publication bias assessed using Egger’s regression showed a significant bias for HR in males (Table 2).

### 3.3. Quality Assessment

Table 3 summarises the outcomes of the risk of bias assessment of the included studies. A total of 15 studies scored a “Low” overall bias, meaning the bias in all measured domains was low. A total of 17 studies were rated with an overall bias of “Some concerns”, meaning that they scored “Some concerns” in one but not more than one of the measured domains. The remaining 31 studies received an overall bias score of “High”. This means that they either scored “High” in at least one of the domains or had some concerns in multiple domains. Other domains that were assessed were “Allocation concealment” to also assess selection bias, “Incomplete outcome data” to assess attrition bias, “Measurement outcomes” to assess detection bias, and, lastly, “Selective reporting” to assess reporting bias.

### 3.4. Systolic Blood Pressure

Data for all parameters are reported as a mean difference and relative change from the baseline in percentages. SBP data are reported in Table 4 and Figure 2. The weighted pre-intervention mean SBP in the female population was 137.3 [133.4; 141.3] mmHg and the weighted pre-intervention mean SBP in the male population was 134.6 [129.8; 139.5] mmHg (*p* = 0.399). SBP decreased in females by 11.1 mmHg (95% CI, −14.5; −7.8) (% change, −7.9% (95% CI, −10.4; −5.4)), as compared to 11.1 mmHg (95% CI, −14.0; −8.2) (% change, −8.2% (95% CI, −10.4; −6.1)) in males; this decrease was not statistically different between sexes (*p* = 0.977). Heterogeneity was high in both female (I^2^ = 85%) and male data (I^2^ = 86%). No clinical factors or methodological sources significantly affected the change in SBP (Table 5).

The mean difference for SBP by treatment duration is reported in Table 6. In females, the observed decrease in SBP is greatest in the sub-acute treatment phase, while the decrease attenuates after prolonged treatment. In males, there is no discernible difference between the different phases (Appendix A and Figure 3).

### 3.5. Diastolic Blood Pressure

The weighted pre-intervention mean DBP in the female population was 85.2 [81.0; 89.3] mmHg and the weighted pre-intervention mean DBP in the male population was 82.8 [78.5; 89.3] mmHg (*p* = 0.444). DBP decreased by 8.0 mmHg (95% CI, −10.6; −5.3) (% change, −9.4% (95% CI, −12.5; −6.2)) in females and by 8.0 mmHg (95% CI, −10.1; −6.0) (% change, −9.7% (95% CI, −12.2; −7.3)) in males (Table 4, Figure 4). The change in DBP between sexes was not statistically significant (*p* = 0.972). Heterogeneity was substantial in female (I^2^ = 83%) and male (I^2^ = 91%) data. There were no sources of heterogeneity that significantly affect the change in DBP (Table 5).

The mean difference for DBP by treatment duration is reported in Table 6. The observed decrease in DBP is more pronounced after sub-acute treatment in males than in females and similar in chronic treatment (Appendix A and Figure 5).

### 3.6. Mean Arterial Pressure

The weighted pre-intervention mean MAP in the female population was 108.8 [101.6; 115.9] mmHg and the weighted pre-intervention mean MAP in the male population was 111.1 [97.8; 124.5] mmHg (*p* = 0.759). MAP decreased 8.1 (95% CI, −11.7; −4.5) (% change, −7.5% (95% CI, −10.9; −4.2)) in females and 9.9 (95% CI, −17.0; −2.8) (% change, −8.9% (95% CI, −10.9; −4.2)) in males, a change comparable in both sexes (Table 4 and Appendix A). The heterogeneity in females was low (I^2^ = 15%), which was contrary to that in males (I^2^ = 92%). The meta-regression analysis showed no clinical or methodological sources of heterogeneity (Table 5).

The mean difference for MAP by treatment duration is reported in Table 6. Chronic treatment showed a similar statistically significant change between females and males (Appendix A and Figure 6). 

### 3.7. Heart Rate

The weighted pre-intervention mean HR in the female population was 76.3 [69.1; 83.4] beats per minute (bpm) and the weighted pre-intervention mean HR in the male population was 74.0 [72.5; 75.6] bpm (*p* = 0.554). The change in HR was comparable between females, −10.8 bpm (95% CI, −17.4; −4.2) (% change, −14.2% (95% CI, −22.8; −5.5)), and males, −9.8 bpm (95% CI, −11.1; −8.4) (% change, −13.2% (95% CI, −15.1; −11.4)) (Table 4, Appendix A), changes were comparable in both sexes (*p* = 0.759). Corrected for publication bias, the mean difference in males was −12.1 bpm (CI, −13.5; −10.7). The heterogeneity was high in females (I^2^ = 98%) as well as in males (I^2^ = 78%). There was no clinical source of heterogeneity. There was, however, one methodological source for heterogeneity, “Low quality” (*p* = 0.027) (Table 5).

The mean difference for HR by treatment duration is reported in Table 6. The observed decrease in HR is similar in sub-acute and chronic treatment for both sexes (Appendix A). 

### 3.8. Cardiac Output

The weighted pre-intervention mean CO in the population was 5.1 [4.8; 5.4] L/min. The decrease in CO in males was 0.1 L/min (95% CI, −0.5; 0.2) (% change, −2.9% (95% CI, −9.2; 3.4)) (Table 4 and Figure 7). This decrease was not statistically significant. For CO, no female data were available and as such, no calculations could be made to compare values between sexes. The effect of metoprolol contributed to the heterogeneity (*p* = 0.044) (Table 5). The mean difference for CO by treatment duration is again reported in Table 6. Data were exclusively available for males, and both acute and chronic treatments showed non-statistically significant changes (Appendix A). 

### 3.9. Left Ventricular Ejection Fraction

The weighted pre-intervention mean LVEF in the female population was 51.8 [41.6; 62.0] %, and the weighted mean LVEF in the male population was 51.2 [43.0; 59.5] % (*p* = 0.926). LVEF in females changed by 4.2% (95% CI, −0.4; 8.8) (% change, 8.0% (95% CI, −0.7; 16.8)) and in males by 3.7% (95% CI, 0.6; 6.9) (% change, 7.2% (95% CI, 1.1; 13.4)) (Table 4, Figure 8). Despite the increase of LVEF in males being statistically significant, the increase in LVEF did not demonstrate a significant difference between sexes (*p* = 0.858). There was a high heterogeneity in both females (I^2^ = 93%) and males (I^2^ = 95%). There were no significant sources of heterogeneity (Table 5).

The mean difference for LVEF by treatment duration is reported in Table 6. Sub-acute data between both sexes was not statistically significant. Chronic treatment, however, showed a similar statistically significant change between females and males (Appendix A).

### 3.10. Left Ventricular Mass

The weighted pre-intervention mean LVM in the population was 283.5 [224.6; 342.5] g. The change in LVM was based on four studies and provided data for males only. There was a decrease in LVM of 20.6 g (95% CI, −56.3; 15.1) (% change, −7.4% (95% CI, −20.2; 5.4)) (Table 4, Appendix A). The heterogeneity was high (I^2^ = 74%). The meta-regression analysis showed three statistically significant methodological sources, respectively “Low quality”, “Medium quality”, and “Percentage of maximum dose” (Table 5).

## 4. Discussion

This systematic review and meta-analysis primarily studied sex differences in effects on hypertension treatment of BBs on BP, HR, and secondary on cardiac geometry and function. BB treatment significantly but comparably lowers BP and HR in both sexes. LVEF increased in both sexes, showing no significant differences between the sexes. However, the rise reached statistical significance only in males. Cardiac geometry did not change appreciably in males and was not reported in females.

BBs are amongst the five classes of antihypertensives recommended to treat hypertension [16]. They significantly reduce the risk of stroke, heart failure, and major cardiovascular events in hypertensive patients [107]. However, sex differences may negatively impact the efficacy of BBs and their protective effects; the anti-ischaemic effect, remaining functional cardiac reserves and survival in patients with hypertension or coronary artery disease using BBs is less in females than in males [26,108,109,110].

Pharmacologically, the mechanism by which BBs exert their effects is primarily by obstructing B1 receptors on the cardiac myocytes, inhibiting the stimulating effect of ligands, epinephrine, and norepinephrine and with it cardiac automaticity and conduction velocity, translating into negative chronotropic and inotropic effects. In the context of the circulatory system, the blockage of juxtaglomerular B1 receptors reduces renin output, resulting in decreased angiotensin-driven vascular tone and lower aldosterone-dependent volume retention. These concurrent modes of action lower BP and reduce cardiac oxygen demand [111]. Oestrogen and progesterone inhibit the cardiac expression of B1-adrenoreceptors, reduce beta-adrenergic mediated stimulation, and are thought to underlie the cardioprotective effects observed in females of reproductive age [112]. In addition, females have up to twice the peak serum concentration and a higher area under the curve regarding metoprolol and propranolol than males [113,114]. This difference is hypothesised to originate from enhanced bowel absorption, smaller distribution volume, and slower clearance via testosterone-affected CYP2D6 metabolisation in females as compared to males [112,114]. Drug exposure in females using oral contraceptives is significantly higher than in females not using oral contraceptives [113,115]. Related to the higher plasma concentration, some studies suggest females exhibit a larger decrease in HR and SBP under BB therapy than males [113,116]. We observed a comparable BP response to BBs in both sexes, while CO does not seem to change significantly. This suggests that BB-induced lower BP is primarily vascular rather than cardiac driven and originates from a reduction in vascular tone, resistance, and circulatory volume instead of a reduction in CO.

Clinically, previous studies comparing the effects of BBs between females and males in several cardiovascular disease states showed conflicting results. On the one hand, a meta-analysis by Olsson et al., including 4353 males and 1121 females on metoprolol therapy after MI, showed a reduction in overall deaths regardless of sex [117]. On the other hand, in heart failure patients, the Metoprolol CR/XL Randomised Intervention Trial in Congestive Heart Failure (MERIT-HF) and the Carvedilol Prospective Randomized Cumulative Survival (COPERNICUS) trial showed, that in contrast to males, the reduction in mortality for females not to be significant [118,119]. In addition, in a study by Fletcher et al., male patients with hypertension treated with BBs had better survival than female patients, which is in line with the findings of the beta-blocker heart attack trial that showed that BB treatment for patients with coronary artery disease was beneficial for males, but not females [108,109]. Nonetheless, in several post hoc analyses of the aforementioned trials, a similar all-cause mortality and all-cause hospitalisation was found between males and females with heart failure with reduced ejection fraction (HFrEF). In the post hoc analysis of the Cardiac Insufficiency Bisoprolol Study II (CIBIS II) study, bisoprolol showed a significant advantage for females over males using bisoprolol for congestive heart failure [120]. Similar results were found in the post hoc analysis of COPERNICUS, MERIT-HF, and CIBIS trials [121,122,123]. In a meta-analysis by Ghali et al., data from the MERIT-HF, COPERNICUS, and CIBIS-II trials were pooled, resulting in similar survival benefits for females and males using BBs with HfrEF [124]. In the Beta-Blocker Evaluation of Survival Trial (BEST), 2708 patients, from which 593 were females and 2115 were males, with a LVEF ≤ 35%, were randomised to bucindolol versus placebo. This study found that within the group of patients that had a nonischaemic aetiology, females had a better survival rate than males. This was the opposite for the group with an ischaemic aetiology. In this group, the males achieved a better survival than females [125]. In a meta-analysis by Shekelle et al., data from BEST, CIBIS-II, COPERNICUS, MERIT-HF, and U.S. Carvedilol were pooled. The meta-analysis shows a significant reduction in mortality for both males and females [126]. 

Several differences may underlie these conflicting results. First, BBs were underused in females with myocardial infarction. Second, females are underrepresented in most trials. Only 25% of the total patients in these five trials were females, leading to reduced study power. Third, included females were older and had more severe clinical conditions than included males [112]. Fourth, it might be that the response and outcome of several different underlying cardiovascular conditions, MI, and heart failure with or without preserved ejection fraction, may be different, and, consequently, affect observed outcomes when combined in the analysis. The most recent meta-analysis (2016) included 13,833 patients from 11 trials of which 24% are female. Included patients had HFrEF and sinus rhythm. This study shows that irrespective of sex, BBs compared with placebo reduce mortality and hospital admissions in these patients [127]. Interestingly, there was an excess of heart failure exacerbations in the youngest age quarter of females compared to males [127]. This finding is interesting, especially in light of a recently published study in which patients presenting with acute coronary syndrome were investigated [110]. In this study, it was observed that females using BBs for hypertension had a higher chance of presenting with heart failure at hospital admission compared to males. 

### Strengths and Limitations

There were several strengths to our systematic review and meta-analysis. To our knowledge, this systematic review and meta-analysis is the first to investigate such an extensive number of studies from inception in a systematic way and aimed to include all relevant studies, which resulted in an analysis of the lack of sex-specific outcome measures in antihypertensive studies. The results of this broad search translated into a large pooled patient population of 2161 patients using BBs and having sex-stratified data available. Another strength was that all articles were screened by two independent reviewers. Furthermore, to detect bias, the Cochrane recommended RoB2 tool was used. 

Our study also has several limitations. First, the quality assessment of the articles showed that half of the included articles were rated as “High risk of bias”. The Risk of Bias 2 (RoB2) tool classifies non-randomised controlled trials as lower than randomised trial. As many of the included studies were prospective non-randomised cohort studies, by using the RoB2 tool, these studies were viewed upon as not within the highest quality and therefore contributing to the risk of bias. However, we cannot rule out an overestimated risk of bias for these studies as the RoB2 tool is developed for RCTs. Second, in many of the included studies, participants received co-medications, next to their study medication. Although we cannot rule out an additional effect on BP control, reported studies observed no significant differences in co-medication between study groups, suggesting comparable effects on both groups. Since most patients commonly use co-medication, this may even contribute to the external validity of our study. Third, only 20.2%of participants were female, lowering the power of findings as compared to males in our study. Many studies did not present their data in a sex-stratified manner. The 20.2%female inclusion rate in our study is not very different from that in large trials, which is around 25% [112]. This may be partly the result of the historical exclusion of pre-menopausal females from clinical trials [128]. Nonetheless, for our primary outcome, we were able to make reliable findings and, therefore, think that the unequal representation of females as compared to males did not affect our results. 

## 5. Conclusions and Recommendations

In summary, beta-blockers comparably lowered blood pressure and heart rate in both sexes. The lack of changes in cardiac output in males suggests that the reduction in blood pressure is predominantly determined by a reduction in vascular resistance. Furthermore, females were underrepresented compared to males. Future research should have a better female-to-male inclusion ratio and sex stratified data when researching treatment effects of hypertensives.

## Figures and Tables

**Figure 1 biomedicines-11-01494-f001:**
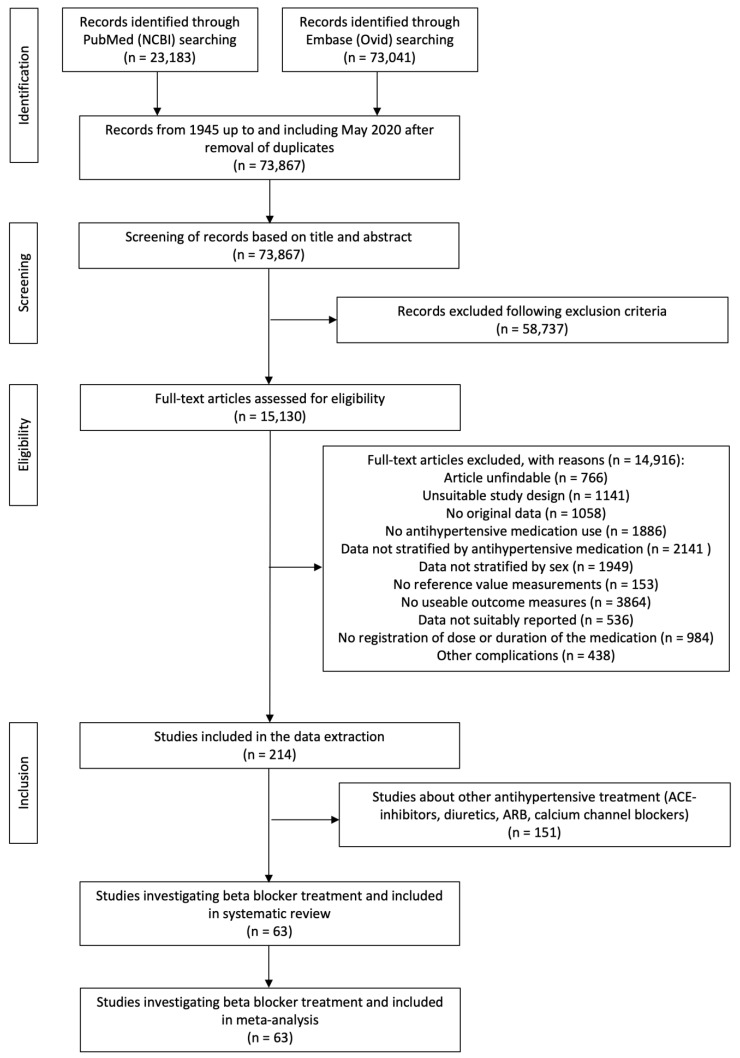
Flowchart of the systematic selection.

**Figure 2 biomedicines-11-01494-f002:**
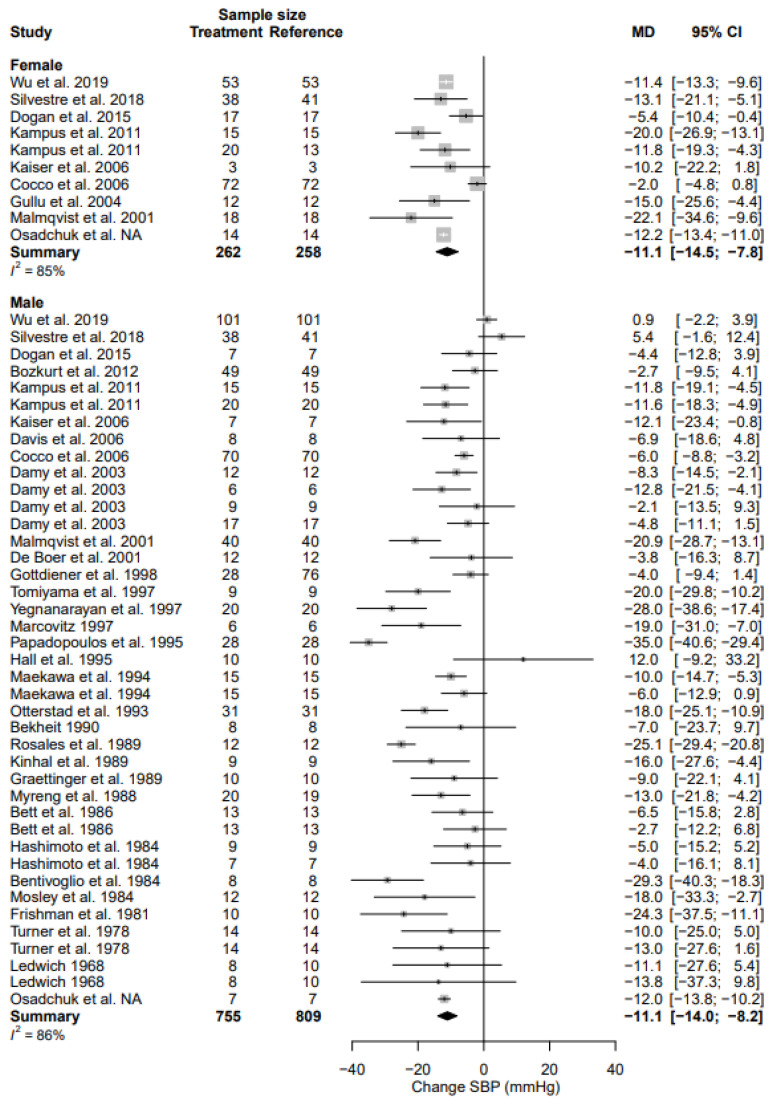
Forest plot of systolic blood pressure (SBP) change in mmHg after beta-blocker use compared to baseline for females and males [24,26,27,28,29,32,34,35,38,41,42,44,45,48,50,54,55,56,57,61,62,63,64,72,73,78,79,82,83,84,86]. MD = mean difference.

**Figure 3 biomedicines-11-01494-f003:**
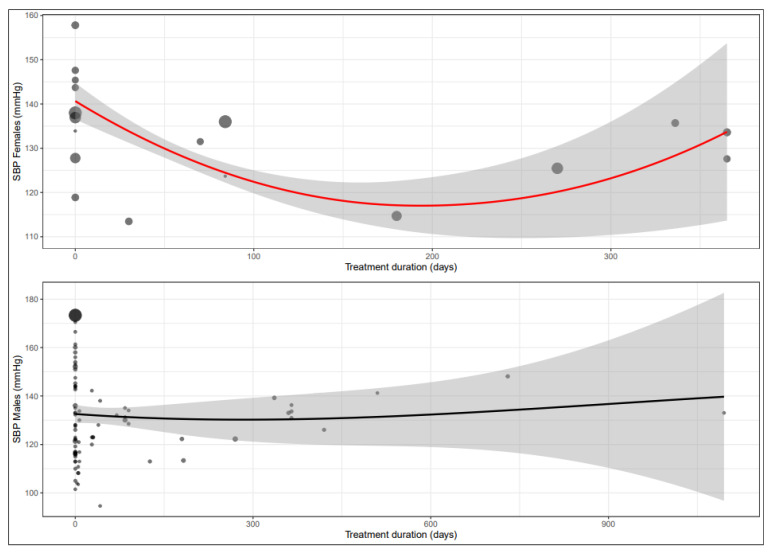
Meta-regression curve of systolic blood pressure (SBP) by beta-blocker treatment duration (days). Every circle represents one article, and the size represents the number of participants included in the study, shown as a small or larger circle.

**Figure 4 biomedicines-11-01494-f004:**
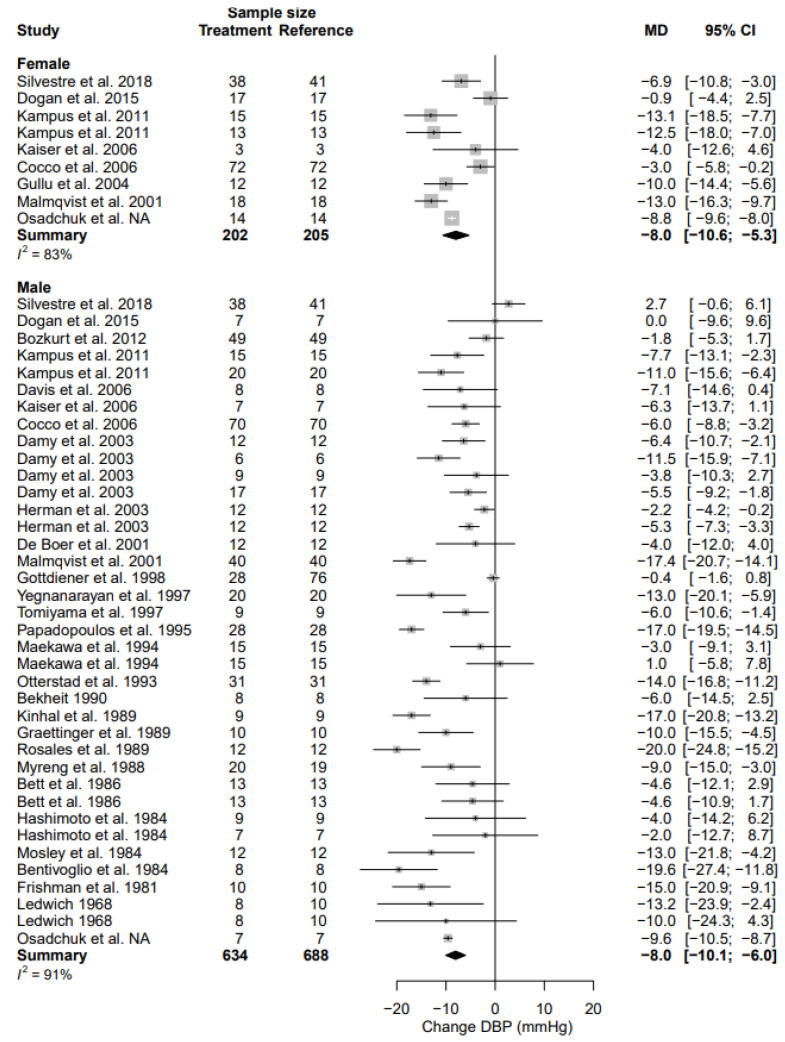
Forest plot of diastolic blood pressure (DBP) change in mmHg after beta-blocker use compared to baseline for females and males [24,26,27,28,32,35,38,42,44,45,46,48,49,50,54,55,56,57,62,63,64,72,73,78,79,82,83,84]. MD = mean difference.

**Figure 5 biomedicines-11-01494-f005:**
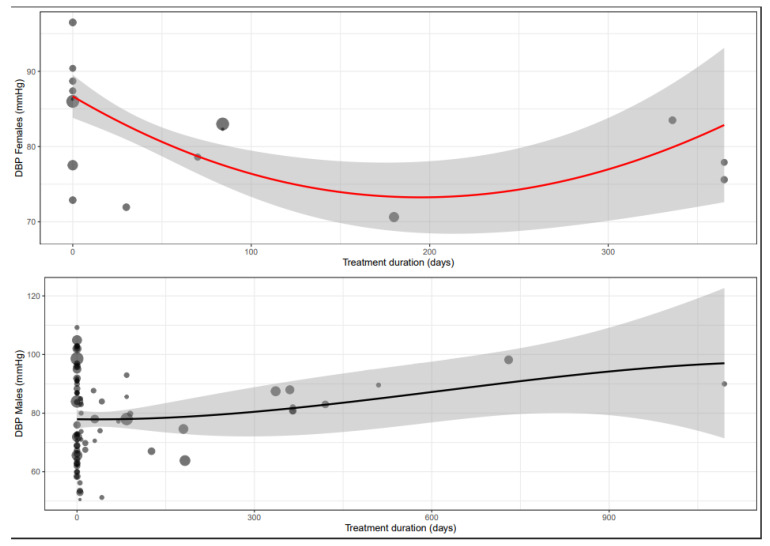
Meta-regression curve of diastolic blood pressure (DBP) by BB treatment duration (days). Every circle represents one article, and the size represents the number of participants included in the study, shown as a small or larger circle.

**Figure 6 biomedicines-11-01494-f006:**
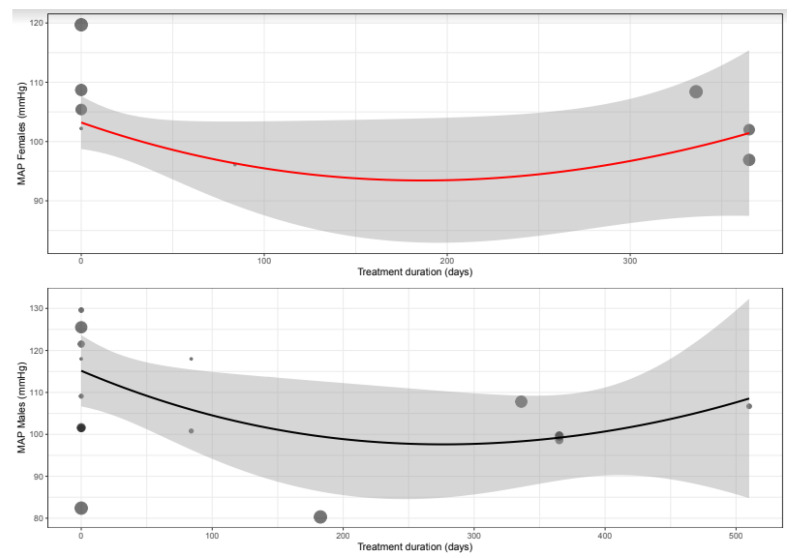
Meta-regression curve of mean arterial pressure (MAP) change in mmHg by BB treatment duration (days). Every circle represents one article, and the size represents the number of participants included in the study, shown as a small or larger circle.

**Figure 7 biomedicines-11-01494-f007:**
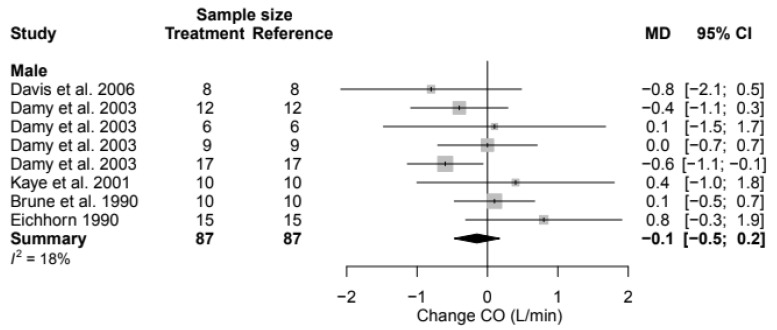
Forest plot of cardiac output (CO) change in L/min after beta-blocker use compared to baseline for males [27,28,66,76,80]. MD = mean difference.

**Figure 8 biomedicines-11-01494-f008:**
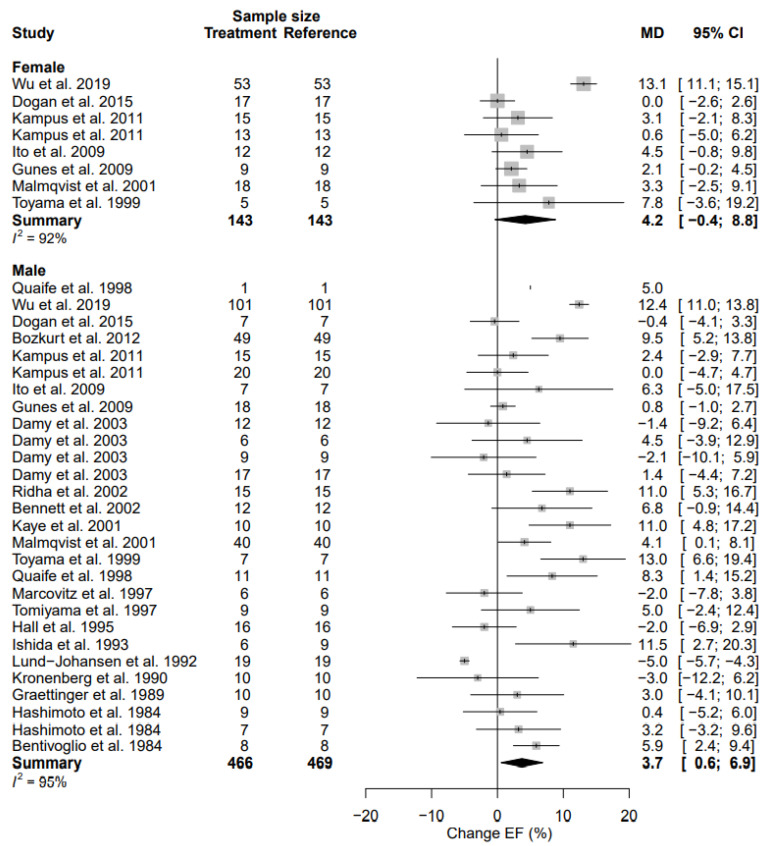
Forest plot of left ventricular ejection fraction (LVEF) change in % after beta-blocker use compared to baseline for females and males [25,27,29,31,32,33,34,40,41,42,45,48,54,63,65,66,67,68,69,72,73,77,84]. MD = mean difference.

**Table 1 biomedicines-11-01494-t001:** Study characteristics (*n* = 63).

Study	Patient	Ethnicity	Beta Blocker Treatment	Mean Dose (mg/Day)	% Max Dose *	Subjects Beta Blockers (*n*)			Control Group **	Controls (*n*)			Age		Intervention Duration (Days)	Study Design	Extracted Variables	Mentioned Methods of Measurement
						Total	M	F		Total	M	F	(Years + SD)					
Bozkurt et al. (2012) [63]	HFrEF	W, B, H	Carvedilol	28.9	57.8	49	49	0	Baseline ***	49	49	0	63.9	9	182.63	PCS	SBP, DBP, MAP, HR, LVEF, LVMI, LVEDVI, LVESVI, E/A ratio, CI, LVESD, LVEDD	Echo
Damy et al. (2003) [27]	He	W	Metoprolol	200	1	36	36	0	Placebo	13	13	0	29.4	4.0	5	RCT	SBP, DBP, HR, CO, LVEF	Echo, ECG, phonocardiogram, carotidogram, TEIC
Kaiser et al. (2006) [78]	T2D, HTN	-	Nebivolol	5	100	10	7	3	Enalapril ***	10	7	3	54.4	8.8	84	RCT	SBP, DBP, MAP, HR	Sphygmomanometer
Kampus et al. (2011) [32]	HTN	-	Nebivolol	5	100	80	41	39	Baseline ***	80	41	39	46.5	9.9	365	RCT	SBP, DBP, MAP, HR, LVEF, LVMI, E/A ratio	Echo, oscillometric
			Metroprolol (succinate)	75	38													
Malmqvist et al. (2001) [48]	HTN, LVH	W	Atenolol	75	75	58	40	18	Irbesartan	56	36	20	54.3	9.5	336	RCT	SBP, DBP, MAP, HR, LVEF, LVMI, E/A ratio, LVEDD,	Sphygmomanometer, echo, ECG
Hashimoto et al. (1984) [84]	HTN	-	Indenolol	30	25	16	16	0	Baseline ***	16	16	0	39.8	10.7	7	PCS	SBP, DBP, HR, LVEF, LVDd, LVDs, CI, MBP	Sphygmomanometer, ECG, echo
Turner et al. (1978) [61]	AP	W	Propranolol or nadolol with placebo	160	50	28	28	0	Baseline ***	28	28	0	53.0	8.0	28	PCS	SBP, HR	Sphygmomanometer, ECG
Bentivoglio et al. (1984) [42]	HTN, LVH	-	Metoprolol (tartrate)	200	50	8	8	0	Baseline ***	8	8	0	37.4	9.0	510	PCS	SBP, DBP, MAP, HR, LVEF	Sphygmomanometer
Papadopoulos et al. (1995) [82]	HTN	-	Bisoprolol	10	50	28	28	0	Baseline ***	28	28	0	52.5	-	30	PCS	SBP, DBP, HR, LVD, LAD	Sphygmomanometer, echo
Gottdiener et al. (1998) [44]	HTN	W, B	Atenolol	62.5	63	76	76	0	Baseline ***	76	76	0	58.8	10.0	56	RCT	SBP, LA	Sphygmomanometer, echo
Tomiyama et al. (1997) [72]	HTN	A	Acebutolol	250	21	9	9	0	Nifedipine	13	13	0	44.8	6.3	1095	RCS	SBP, DBP, LVEF, LVMI, RWTd	Sphygmomanometer, echo
Yegnanarayan et al. (1997) [62]	HTN, LVH	-	Propranolol	80	25	20	20	0	Abana	20	20	0	-	-	420	RCT	SBP, DBP, IVST, LVPWT, FS, LVIDs, LVIDd	Sphygmomanometer, echo
Maekawa et al. (1994) [56]	He	-	Propranolol	20	6	15	15	0	Baseline ***	15	15	0	-	-	0.08	PCS	SBP, DBP, HR	Sphygmomanometer, echo, ECG
			Nipradilol	6	33													
De Boer et al. (2001) [64]	HFrEF	-	Carvedilol	61.5	123	12	12	0	Placebo	5	5	0	60.5	8.2	90	RCT	SBP, DBP, HR	-
Otterstad et al. (1993) [50]	HTN	-	Atenolol	75	75	31	31	0	Co-amiloride	49	48	1	45.9	11.5	360	RCT	SBP, DBP, HR, LVM, LVMI, IVST, LVPWT, RWT, LVID	ECG, echo
Ledwich (1968) [55]	AMI	-	Propranolol	60 and 120	19 and 38	20	20	0	Placebo	20	20	0	60.4	8.0	7	RCT	SBP, DBP, HR	ECG
Bett et al. (1986) [79]	He	-	Bucindolol	50 and 200	25 and 100	13	13	0	Placebo ***	13	13	0	22.0	1.0	0.167	RCT	SBP, DBP, HR, CI, FS	Sphygmomanometer, echo, ECG
Frishman et al. (1981) [73]	HTN, AP	-	Labetalol	1050	44	10	10	0	Baseline ***	10	10	0	59.4	5.78	28	PCS	SBP, DBP, HR, LVEF, LVEDVI, LVESVI, CI, IVSTd, LVPWT	Sphygmomanometer, ECG, echo
Davis et al. (2006) [28]	HFrEF	-	Metoprolol (succinate)	118.75	59	8	8	0	Unchanged medication	8	8	0	64.2	2.5	42	RCT	SBP, DBP, HR, CO	ECG, oscillometric
Dogan et al. (2015) [54]	Migraine	-	Propranolol	80	25	24	7	17	No medication	80	25	55	33.3	9.6	30	PCS	SBP, DBP, HR, LVEF, IVST, LVPWT, LVESD, LVEDD	Echo, oscillometric
Myreng et al. (1988) [49]	AP	-	Atenolol	100	100	20	20	0	Healthy subjects	18	16	2	-	-	126	PCS	SBP, DBP, HR, desceleraton, E/A ratio, E, A, SV, IVRT	echo, ECG
Cocco et al. (2006) [26]	AP	-	Metoprolol (succinate)	119	60	142	70	72	Baseline ***	142	70	72	58.0	9.2	84	RCT	SBP, DBP, HR	ECG
Bekheit (1990) [24]	MI	-	Metoprolol	200	100	8	8	0	Diltiazem, nifedipine	19	19	0	62.0	13.0	6	PCS	SBP, DBP, HR	Sphygmomanometer, ECG
Rosales et al. (1989) [86]	HTN, LVH	-	Carteolol	14.8	25	16	16	0	Baseline ***	16	16	0	58.9	4.0	365	RCS	SBP, DBP, MAP	ECG
Marcovitz (1997) [34]	He	-	Metoprolol	50	25	6	6	0	Baseline ***	6	6	0	-	-	3	PCS	SBP, HR, LVEF, FS	ECG
Hall et al. (1995) [29]	HF	B, W	Metoprolol	56.25	28	16	16	0	Standard Therapy	10	10	0	54.0	3.6	90	RCT	SBP, HR, LVEF, LVM, LVEDV, LVESV	Echo
Kinhal et al. (1989) [83]	HTN, HFrEF	B, W	Dilevalol	400	25	9	9	0	Baseline ***	9	9	0	60.0	-	39	PCS	SBP, DBP, HR	Sphygmomanometer, ECG
Mosley et al. (1984) [57]	HTN	B, W	Propranolol	167.5	52	12	12	0	Guanabenz	14	14	0	48.2	4.8	42	RCT	SBP, DBP, HR, CO, LVM, SVR	Sphygmomanometer, echo
Graettinger et al. (1989) [45]	HTN	W	Atenolol	156	156	10	10	0	Lisinopril	9	9	0	56.0	-	84	RCT	SBP, DBP, HR, LVEF, LVM, IVST, LVPWT, LVIDd, RWT, RVD	Sphygmomanometer, echo
Silvestre et al. (2018) [38]	Cirrhotic cardiomyopathy	W, O	Metoprolol (succinate)	120	60	41	18	23	Placebo	37	14	23	50.4	-	180	RCT	SBP, DBP, HR	Echo
Wu et al. (2019) [41]	CHF	A	Metoprolol	99.75	50	154	101	53	Baseline ***	154	101	53	66.4	-	270	PCS	SBP, HR, LVEF, CI	Sphygmomanometer, echo, ECG
Osadchuk et al. (2019) [35]	HTN, CHD	-	Metoprolol (succinate)	56.1	28	21	7	14	Ramipril	20	8	12	70.6	7.2	70	RCT	SBP, DBP, HR	Oscillometric, ECG
Herman et al. (2003) [46]	He	-	Carvedilol	18.75	38	12	12	0	Baseline ***	12	12	0	21.6	0.3	14	RCT	DBP, HR	Sphygmomanometer
			Atenolol	37.5	38													
Zemel et al. (1990) [52]	HTN, LVH	B	Atenolol	50	50	6	6	0	Calcium supplements ***	6	6	0	-	-	84	CCS	MAP, E/A ratio, LVPWT, FS	Oscillometric, ECG, echo
Ridha et al. (2002) [69]	CHF	W, B	Carvedilol	32	64	15	15	0	Baseline ***	15	15	0	62.0	11.0	84	PCS	HR, LVEF, MBP	Oscillometric, ECG
Silke et al. (1997) [37]	He	-	Metoprolol	50	25	9	9	0	Placebo ***	9	9	0	22.1	-	0.33	RCT	HR	ECG
			Celiprolol	200	50													
Silke et al. (1986) [58]	AP, MI	-	Propranolol (i.v.)	8	2.5	32	32	0	Baseline ***	32	32	0	52.8	7.0	0.0087	PCS	HR, CI, SVRI	Catheter
			Pindolol (i.v.)	0.8	11													
Silke et al. (1985) [71]	MI	-	Acebutolol (i.v.)	25 and 50	4 and 8	24	24	0	Baseline ***	24	24	0	45.0	-	0.17	RCT	HR, SVRI, CI	ECG, catheter
			Acebutolol	200 and 400	16 and 32													
Silke et al. (1984) [59]	MI	-	Propranolol (i.v.)	8	12	16	16	0	Baseline ***	16	16	0	54.0	1.8	0.01	PCS	HR, SVRI, CI	Catheter
			Labetalol (i.v.)	40	14													
Silke et al. (1984) [60]	AP, MI	-	Propranolol (i.v.)	8	8	16	16	0	Baseline ***	16	16	0	51.5	3.3	0.00868	PCS	HR, SVRI, CI	Catheter
Silke et al. (1984) [81]	AP, MI	-	Propranolol (i.v.)	0.8	11	12	12	0	Baseline ***	12	12	0	52.8	5.5	0.0087	PCS	HR, CI, SVRI	Catheter
			Pindolol (i.v.)	8	11													
Taniguchi et al. (2003) [39]	CHF	-	Metoprolol	30 and 60	15 and 30	12	10	2	Baseline ***	12	10	2	54.0	12.0	432	PCS	HR, MBP	Echo
			Atenolol	25	25													
			Carteolol	10 and 20	17 and 33													
Kyriakides et al. (1992) [47]	AMI	-	Atenolol (i.v.)	5	5	28	28	0	Baseline ***	28	28	0	53.0	6.0	0.0069	PCS	HR, CI, MBP, SVR	ECG, catheter
Bennett et al. (2002) [25]	CHF	-	Metoprolol (succinate)	106.25	53	12	12	0	Baseline ***	12	12	0	62.0	10.0	180	PCS	HR, LVEF	PET
Brune et al. (1990) [76]	CHD	-	Nebivolol	5	100	10	10	0	Baseline ***	10	10	0	56.7	4.8	7	PCS	HR, CO, RAP, SV	Catheter
Renard et al. (1983) [36]	MI	-	Metoprolol (tartrate)	150	38	9	9	0	Baseline ***	9	9	0	53.0	-	1.04	PCS	HR	Catheter
Renard et al. (1984) [75]	AMI, HTN	-	Labetalol (i.v.)	126	5	18	18	0	Baseline ***	18	18	0	56.8	-	0.04	PCS	HR, CI	Catheter
Kaye et al. (2001) [66]	HF	-	Carvedilol	42.5	85	10	10	0	Baseline ***	10	10	0	55	3.0	90	PCS	HR, CO, LVEF, RVEF	Catheter
Frais et al. (1985) [43]	AMI	-	Atenolol (i.v.)	8	-	16	16	0	Baseline	16	16	0	-	-	0.0087	RCT	HR, SVRI, CI	ECG, catheter
			Acebutolol (i.v.)	80	-													
Aronow et al. (1975) [53]	AP, CHD	-	Tolamolol	10 and 20	1 and 2	45	45	0	Saline	15	15	0	51.1	-	0.0035	RCT	HR	ECG
			Propranolol	10	3													
Heesch et al. (1995) [30]	HF	-	Metoprolol	100	50	30	30	0	Baseline ***	30	30	0	48.0	11.0	90	RCT	HR, CI, LVEDP, LVESP	Catheter
			Bucindolol	175	88													
Ishida et al. (1993) [31]	CHF	-	Metoprolol	45.6	23	9	9	0	Baseline ***	9	9	0	52.6	10.7	180	PCS	HR, LVEF, LVESD, LVEDD, FS	Echo
Todd et al. (1990) [51]	AP	-	Atenolol	100	100	20	20	0	No beta-blocker	20	20	0	52.0	-	7	RCT	HR	ECG
Kronenberg et al. (1990) [33]	AMI	-	Metoprolol (i.v.)	12.5	19	10	10	0	Normal subjects	13	-	-	42.1	14.2	0.0069	PCS	HR, LVEF, RVEF, LVEDV, LVESV, SV	Sphygmomanometer, catheter, radionuclide studies
Nelson et al. (1983) [74]	AMI	-	Labetalol (i.v.)	242	10	21	21	0	Baseline ***	21	21	0	53.0	1.5	0.1146	PCS	HR, CI, SVRI	ECG, catheter
Littler (1985) [85]	HTN	-	Timolol	30	50	9	9	0	Nifedipine, indapamide	17	11	6	39.7	10.6	112	PCS	HR, LVMI	Echo,
Yeoh et al. (2011) [70]	Early DCM	-	Carvedilol	15.63	31	16	9	7	Placebo	16	8	8	39.5	11.3	180	RCT	HR, E, A, E/A ratio, LVESD, LVEDD, FS	Echo
Ito et al. (2009) [65]	Idiopathic DCM	-	Carvedilol	7.4	15	19	7	12	Baseline ***	19	7	12	47.9	10.3	60	PCS	HR, LVEF, E/A ratio, LVEDD, DT, Ea mean	Echo
Eichhorn et al. (1990) [80]	HFrEF	-	Bucindolol	175	88	15	15	0	Baseline ***	15	15	0	50.0	11.0	90	PCS	HR, CO, SV, LVEDP, SVR	Catheter
Toyama et al. (1999) [40]	DCM, HF	-	Metoprolol	31.25	0.16	12	7	5	Enalapril ***	12	7	5	58.0	12.0	365	PCS	LVEF, LVESD, LVEDD	Echo
Quaife et al. (1998) [68]	HFrEF	-	Carvedilol	56.25	1.13	11	10	1	Placebo	11	10	1	53.5	11.8	120	RCT	LVEF, RVEF	Radio-nuclide ventriculography, catheter
Gunes et al. (2009) [77]	CSF	-	Nebivolol	5	1	27	18	9	Subjects without CSF	27	16	11	54.7	10.9	90	PCS	LVEF, LVEDV, LVESV, LVEDD, DT, IVRT	Echo
Lund-Johansen et al. (1992) [67]	HTN	-	Carvedilol	62	1.24	19	19	0	Baseline ***	19	19	0	44.0	-	224	PCS	LVEF, E, A, E/A ratio, IVST, LVPWT, LVESD, LVEDD, FS	Echo

Data are presented as mean ± SD or percentages. T2D = type 2 diabetes mellitus, HTN = hypertension, HFpEF = heart failure preserved ejection fraction, HFrEF = heart failure reduced ejection fraction, CHD = coronary heart disease, DCM = dilated cardiomyopathy, CKD = chronic kidney disease, (A)MI = (acute) myocardial infarction, LVH = left ventricular hypertrophy, AP = angina pectoris, (C)HF = (chronic) heart failure, CSF = coronary slow flow, He = healthy, W = white, B = black, i.v. = intravenous, SD = standard deviation, RCT = randomised controlled trial, SBP = systolic blood pressure, DBP = diastolic blood pressure, HR = heart rate, CO = cardiac output, LVEF = left ventricular ejection fraction, LVM = left ventricular mass, ECG = electrocardiography, and echo = echocardiography. * Percentage of maximal dosage for the indication hypertension. Metoprolol 400 mg/day orally [87]; Labetolol 2400 mg/day orally [88] and 300 mg/day i.v. [89]; Atenolol 100 mg/day orally [90]; Carvedilol 50 mg/day orally [91]; Acebutolol 1200 mg/day orally [92]; Bisoprolol 20 mg/day orally [93]; Pindolol 30 mg/day orally [94] and 7.2 mg/day i.v. [58]; Nebivolol 5 mg/day orally [95]; Indenolol 120 mg/day orally [96]; Propranolol 320 mg/day orally [97] and 72 mg/day i.v. [98]; Nadolol 320 mg/day orally [99]; Nipradilol 18 mg/day orally [100]; Bucindolol 200 mg/day orally [101]; Carteolol 60 mg/day orally [102]; Dilevalol 1600 mg/day orally [103]; Tolamolol 900 mg/day orally [104]; Celiprolol 400 mg/day orally [105]; Timolol 60 mg/day orally [106]. ** Control group: other antihypertensive treatment, placebo, or non-drug intervention. *** SD not reported.

**Table 2 biomedicines-11-01494-t002:** Publication bias using Egger’s regression for all variables.

	Male	Female	cMD Male	cMD Female
DBP	0.4947	0.7065		
SBP	0.5928	0.9189		
MAP	0.3027	0.8006		
CO	0.3867	-		
HR	0.0341	0.7028	−12.1 [−13.5; −10.7]	
LVEF	0.266	0.4353		
LVM	0.1025	-		

cMD = corrected mean difference.

**Table 3 biomedicines-11-01494-t003:** Quality assessment.

	Random Sequence Allocation (Selection Bias)	Allocation Concealment (Selection Bias)	Incomplete Outcome Data (Attrition Bias)	Measurements Outcomes (Detection Bias)	Selective Reporting (Reporting Bias)	Overall Bias
Bozkurt et al. (2012) [63]	High	Low	Low	Low	Low	High
Damy et al. (2003) [27]	Low	Low	Low	Low	Low	Low
Kaiser et al. (2006) [78]	Low	Low	Low	Low	Low	Low
Kampus et al. (2011) [32]	Low	High	Low	Low	Some concerns	High
Malmqvist et al. (2001) [48]	Low	Low	Low	Low	Low	Low
Hashimoto et al. (1984) [84]	High	Some concerns	Low	Low	Low	High
Turner et al. (1978) [61]	Low	Low	Low	Low	Low	Low
Bentivoglio et al. (1984) [42]	High	Low	Low	Low	Low	High
Papadopoulos et al. (1995) [82]	High	Some concerns	Low	Low	Low	High
Gottdiener et al. (1998) [44]	Low	Low	Low	Low	Low	Low
Tomiyama et al. (1997) [72]	Some concerns	Low	Low	Low	Low	Some concerns
Yegnanarayan et al. (1997) [62]	Some concerns	Low	Low	Low	Low	Some concerns
Maekawa et al. (1994) [56]	High	Low	Low	Low	Low	High
De Boer et al. (2001) [64]	Low	Low	Low	Low	Low	Low
Otterstad et al. (1993) [50]	Low	Low	Low	Low	Low	Low
Ledwich (1968) [55]	High	Low	Low	Low	Some concerns	High
Bett et al. (1986) [79]	Some concerns	Low	Low	Low	Low	Some concerns
Frishman et al. (1981) [73]	Some concerns	Low	Low	Low	Low	Some concerns
Davis et al. (2006) [28]	Low	Some concerns	Low	Low	Low	Some concerns
Dogan et al. (2015) [54]	Some concerns	Low	Low	Low	Some concerns	Some concerns
Myreng et al. (1988) [49]	High	Some concerns	Low	Low	Low	High
Cocco et al. (2006) [26]	Some concerns	Low	Low	Low	Low	Some concerns
Bekheit (1990) [24]	High	Low	Low	Low	Low	High
Rosales et al. (1989) [86]	High	Some concerns	Low	Low	Low	High
Marcovitz (1997) [34]	High	Low	Low	Some concerns	Low	High
Hall et al. (1995) [29]	Some concerns	Low	Low	Low	Low	Some concerns
Kinhal et al. (1989) [83]	High	Low	Low	Low	Low	High
Mosley et al. (1984) [57]	Some concerns	High	High	Low	Low	High
Graettinger et al. (1989) [45]	Low	Low	High	Low	Low	High
Silvestre et al. (2018) [38]	Low	Low	Low	Low	Low	Low
Wu et al. (2019) [41]	High	Low	Low	Low	Low	High
Osadchuk et al. (2019) [35]	Low	Low	Low	Low	Low	Low
Herman et al. (2003) [46]	Some concerns	Low	Low	Low	Low	Some concerns
Zemel et al. (1990) [52]	High	Low	Low	Low	Low	High
Ridha et al. (2002) [69]	High	Low	Low	Low	Low	High
Silke et al. (1997) [37]	Low	Low	Low	Low	Low	Low
Silke et al. (1986) [58]	Low	Low	Low	Low	Low	Low
Silke et al. (1985) [71]	Some concerns	Low	Low	Low	Low	Some concerns
Silke et al. (1984) [59]	Low	Some concerns	Low	Low	Low	Some concerns
Silke et al. (1984) [60]	High	Low	Low	Low	Low	High
Silke et al. (1984) [81]	High	Low	Low	Low	Low	High
Taniguchi et al. (2003) [39]	High	Low	Low	Low	Low	High
Kyriakides et al. (1992) [47]	High	Low	Low	Low	Low	High
Bennett et al. (2002) [25]	High	Low	Low	Low	Low	High
Brune et al. (1990) [76]	High	Low	Low	Low	Low	High
Renard et al. (1983) [36]	Some concerns	Low	Low	Low	Low	Some concerns
Renard et al. (1984) [75]	High	Some concerns	Low	Low	Low	High
Kaye et al. (2001) [66]	Some concerns	Low	Low	Low	Low	Some concerns
Frais et al. (1985) [43]	Low	Low	Low	Low	Low	Low
Aronow et al. (1975) [53]	Low	Low	Low	Low	Some concerns	Some concerns
Heesch et al. (1995) [30]	Low	Low	Low	Low	Low	Low
Ishida et al. (1993) [31]	High	Some concerns	Low	Low	Low	High
Todd et al. (1990) [51]	High	Low	High	Low	Low	High
Kronenberg et al. (1990) [33]	High	Low	Low	Low	Low	High
Nelson et al. (1983) [74]	Some concerns	Low	Low	Low	Low	Some concerns
Littler (1985) [85]	Some concerns	Low	Low	Low	Low	Some concerns
Yeoh et al. (2011) [70]	Low	Low	Low	Low	Low	Low
Ito et al. (2009) [65]	High	Low	Low	Low	Low	High
Eichhorn et al. (1990) [80]	High	Low	Low	Low	Low	High
Toyama et al. (1999) [40]	Some Concerns	Low	Low	Low	Low	Some Concerns
Quaife et al. (1998) [68]	Low	Low	Low	Low	Low	Low
Gunes et al. (2009) [77]	High	Low	Low	Low	Low	High
Lund-Johansen et al. (1992) [67]	High	Low	Low	Low	Low	High

The scores ‘low’, ‘some concerns’ and ‘high’, have a green, yellow, and red background respectively.

**Table 4 biomedicines-11-01494-t004:** Pooled changes induced by beta-blockers in cardiovascular and haemodynamic variables for females and males.

Variable		Females	I^2^	Males	I^2^	*p*
SBP	MD	−11.1 (95% CI, −14.5; −7.8)	85%	−11.1 (95% CI, −14.0; −8.2)	86%	0.9767
	%	−7.9% (95% CI, −10.4; −5.4)		−8.2% (95% CI, −10.4; −6.1)		
DBP	MD	−8.0 (95% CI, −10.6; −5.3)	83%	−8.0 (95% CI, −10.1; −6.0)	91%	0.9715
	%	−9.4% (95% CI, −12.5; −6.2)		−9.7% (95% CI, −12.2; −7.3)		
MAP	MD	−8.1 (95% CI, −11.7; −4.5)	15%	−9.9 (95% CI, −17.0; −2.8)	92%	0.6594
	%	−7.5% (95% CI, −10.9; −4.2)		−8.9% (95% CI, −10.9; −4.2)		
HR	MD	−10.8 (95% CI, −17.4; −4.2)	98%	−9.8 (95% CI, −11.1; −8.4)	78%	0.7585
	%	−14.2% (95% CI, −22.8; −5.5)		−13.2% (95% CI, −15.1; −11.4)		
CO	MD	N.A.	N.A.	−0.1 (95% CI, −0.5; 0.2)	18%	N.A.
	%	N.A.		−2.9% (95% CI, −9.2; 3.4)		
LVEF	MD	4.2 (95% CI, −0.4; 8.8)	92%	3.7 (95% CI, 0.6; 6.9)	95%	0.8583
	%	8.0% (95% CI, −0.7; 16.8)		7.2% (95% CI, 1.1; 13.4)		
LVM	MD	N.A	N.A.	−20.6 (95% CI, −56.3; 15.1)	74%	N.A.
	%	N.A.		−7.4% (95% CI, −20.2; 5.4)		

**Table 5 biomedicines-11-01494-t005:** *p*-Values of meta-regression analysis.

Sources of Heterogeneity	SBP	DBP	HR	CO	LVEF	MAP	LVM
Atenolol	0.7715	0.3898	0.8726	-	0.8072	-	-
Bisoprolol	0.1544	0.1912	0.9143	-	-	-	-
Bucindolol	0.5205	0.6509	0.4296	-	-	-	-
Carteolol	0.6041	0.1132	-	-	-	0.2227	-
Carvedilol	0.2460	0.9885	0.7797	0.6611	0.9052	0.3838	-
Celiprolol	-	-	0.4035				
Dilevalol	0.7909	0.0744	0.4172	-	-	-	-
Indenolol	0.3395	0.9763	0.7211	-	0.7071	-	-
Labetalol	0.3795	0.1292	0.8784	-	0.6314	-	-
Metoprolol	0.4296	0.5674	0.6810	0.0442	0.8987	0.9291	0.2770
Nadolol	0.9331	-	0.2502	-	-	-	-
Nebivolol	0.7113	0.6799	0.6141	0.2718	0.6424	0.6786	-
Nipradilol	0.6047	0.9661	0.3412	-	-	-	-
Pindolol	-	-	0.5646	-	-	-	-
Propanolol	0.6904	0.6387	0.5360	-	0.4853	-	-
Timolol	-	-	0.6154	-	-	-	-
Tolamolol			0.4957	-	-	-	-
Low quality	0.3656	0.1203	0.0465	0.0545	0.7036	0.6320	0.0072
Medium quality	0.1475	0.1570	0.3808	0.3622	0.9941	-	0.0063
Maximum treatment duration	0.3398	0.3340	0.6555	0.0996	0.3549	0.0584	0.4968
Percentage of maximum dose	0.9874	0.6394	0.5207	0.8426	0.3179	0.7050	0.0010

**Table 6 biomedicines-11-01494-t006:** Pooled changes in cardiovascular and haemodynamic parameters by beta-blocker treatment duration for females and males.

Variable		Females	Males
SBP (mmHg)	MD acute MD sub-acute MD chronic	N.A. −5.4 (−10.4; −0.4) −11.7 (−15.4; −8.0)	−7.8 (−10.0; −5.5) −17.7 (−32.3; −3.1) −11.5 (−25.4; −7.7)
DBP (mmHg)	MD acute MD sub-acute MD chronic	N.A. −0.9 (−25.4; 2.5) −8.9 (−11.4; −6.5)	−5.0 (−6.7; −3.3) −12.0 (−19.7; −4.3) −9.4 (−12.3; −6.4)
HR (bpm)	MD acute MD sub-acute MD chronic	N.A. −11.8 (−16.5; −7.1) −10.7 (−17.8; −3.7)	−8.2 (−10.1; −6.4) −13.1 (−15.7; −10.6) −11.0 (−13.2; −8.8)
CO (L/min)	MD acute MD sub-acute MD chronic	N.A. N.A. N.A.	−0.2 (−0.5; 0.1) N.A. 0.2 (−0.8; 1.1)
LVEF (%)	MD acute MD sub-acute MD chronic	N.A. 0.0 (−2.6; 2.6) 4.9 (0.0; 9.9)	0.3 (−2.1; 2.7) −0.4 (−3.9; 3.1) 5.6 (1.4; 9.9)

## Data Availability

No individual patient data are included in this study. Search strategy and results of included papers are presented within the manuscript and are available upon request.

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
