# Peer review of "Exploring Sex Differences of Beta-Blockers in the Treatment of Hypertension: A Systematic Review and Meta-Analysis"

_biomedicines, 2023, doi:10.3390/biomedicines11051494_

Round 1

Reviewer 1 Report

This interesting systematic review and meta-analysis by Wilmes et al explored sex differences of beta-blockers in the treatment of hypertension. However, I have a few notes:

- page 4 line 15: the current guidelines are missing

- how many articles in the Dutch language were analyzed, with which population? this could make the study less reproducible

- A total of 63 articles were included, for a total of 2052 participants. Among them, only 414 (20.2%) of participants were female, which as pointed out by the authors, reduces the power of findings as compared to males.

- for some parameters such as CO and LVM there were not even data on the female population and perhaps these results should not have been shown because they did not reflect the initial objective of the article.

Reviewer 2 Report

This is a nice systematic review and meta-analysis of beta-blockers in the treatment of hypertension. However, language and grammar errors should be checked. Here are the other two comments: 

1. Line 14, why "beta-blockers are considered amongst the first-line drug treatment options" in the current manuscript?

2. Line 23, what are the five major groups of antihypertensive drugs and the relations with the beta-blockers? Authors should consider discussing them in the Introduction.

Language and grammar errors should be checked.
